# Impact of Ed-LinQ: A Public Policy Strategy to Facilitate Engagement between Schools and the Mental Health Care System in Queensland, Australia

**DOI:** 10.3390/ijerph18157924

**Published:** 2021-07-27

**Authors:** Luis Salvador-Carulla, Ana Fernandez, Haribondhu Sarma, John Mendoza, Marion Wands, Coralie Gandre, Karine Chevreul, Sue Lukersmith

**Affiliations:** 1Centre for Mental Health Research, Australian National University, Canberra 2601, Australia; luis.salvador-carulla@anu.edu.au (L.S.-C.); haribondhu.sarma@anu.edu.au (H.S.); 2Menzies Centre for Health Policy, The University of Sydney, Sydney 2006, Australia; afernand@aspb.cat; 3Agencia de Salut Publica, 08023 Barcelona, Spain; 4ConNectica Consulting, Caloundra 4551, Australia; jmendoza@connetica.com.au (J.M.); mwands@connetica.com.au (M.W.); 5IRDES, Institut de Recherche et Documentation en Économie de La Santé, 75019 Paris, France; gandre@irdes.fr; 6ECEVE, INSERM 1123, 75010 Paris, France; karine.chevreul@inserm.fr

**Keywords:** child and adolescent, mental health policy, mental health care provision, education sector, early intervention, quality, Adoption Impact Ladder, impact analysis, ecosystem

## Abstract

Ed-LinQ is a mental health policy initiative to enhance the early detection and treatment of children with mental illness by improving the liaison between schools and health services in Queensland, Australia. We measured its impact from policy to practice to inform further program developments and public strategies. We followed a mixed quantitative/qualitative approach. The Adoption Impact Ladder (AIL) was used to analyse the adoption of this initiative by end-users (decision makers both in the health and education sectors) and the penetration of the initiative in the school sector. Survey respondents included representatives of schools (*n* = 186) and mental health providers (*n* = 78). In total, 63% of the school representative respondents were at least aware of the existence of the Ed-LinQ initiative, 74% were satisfied with the initiative and 28% of the respondent schools adopted the initiative to a significant extent. Adoption was higher in urban districts and in the health sector. The overall level of penetration in the school sector of Queensland was low (3%). The qualitative analysis indicated an improvement in the referral and communication processes between schools and the health sectors and the importance of funding in the implementation of the initiative. Mapping of existing programs is needed to assess the implementation of a new one as well as the design of different implementation strategies for urban and rural areas. Assessing the adoption of health policy strategies and their penetration in a target audience is critical to understand their proportional impacts across a defined ecosystem and constitutes a necessary preliminary step for the evaluation of their quality and efficiency.

## 1. Introduction

The World Health Organization called for a higher focus on policy and global action in child and adolescent mental health (CAMH) [1], and subsequent international recommendations have been developed to strengthen their implementation nationally and locally [2,3]. The focus on CAMH should become a major priority worldwide as a considerable proportion of mental health problems experienced by adults originates early in life [4,5]. In Australia, surveys report that over 14% of children and young people experience clinically significant mental health problems each year [6]. In Queensland, this estimate was 15.4% for children under 14 years old and 19.8% for young people between 15 and 24 years old in 2013 [7].

This epidemiological burden is likely to increase over time as childhood mental disorders are expected to rise and neuropsychiatric disorders are assumed to cause significant adjustment issues in youth and young people [8]. Concurrently, a growing body of evidence suggests that the opportunities for preventing mental ill-health are greatest when directed at children and young people and that early intervention strategies can be effective in delaying the onset of these disorders as well as in alleviating their collateral damage [9,10,11,12].

School-based programs offer opportunities to reach all children, including at-risk groups and children with early symptoms of mental disorders [3,13]. School staff are often the first port of call for young people who are experiencing mental health difficulties and tend to be one of the first groups outside the family to notice problems. They require access to resources, support, and referral options concomitantly to measures to integrate the education and mental health systems [14].

In this context, the state of Queensland implemented a policy initiative aimed at enhancing the early detection and treatment of mental illness in school-aged children and young people by facilitating communication between schools, the primary care sector and mental health professionals [15]. Queensland has a population of over 5.1 million, concentrated along the eastern coastline and particularly in southeastern Queensland in and around the state capital Brisbane. The proportion of the Queensland population that lives outside of a major city in a regional or remote area is 38% [16].

The program, called the Ed-LinQ initiative and funded by the Queensland Plan for Mental Health 2007–2017, was implemented in twelve districts across the state from 2009 onwards. It was led by the central policy unit of Queensland Health and developed in partnership with government, independent and catholic school systems, and the peak body for general practice, General Practice Queensland. Planning commenced in 2007 and the initiative was informed on the available evidence for school-based mental health interventions. A ‘Framework for Action’ was developed by stakeholders at the strategic level to provide a consistent state-wide approach and a basis for collaborative interdepartmental and interagency relationships. This includes planning and governance mechanisms and interagency memorandums of understanding (MOU). To enhance capacity, there were joint workforce development strategies developed, clinical guidance, consultation liaison protocols for district Ed-LinQ coordinators and child and youth mental health information for distribution to stakeholders. At an operational level, the Framework for Action aimed to provide governance and guidance to stakeholders in mental health, primary health care and education staff, training with respect to identifying at-risk students, information and resources for referral pathways, establishment of management groups, regular meetings, clear processes and others.

Ed-LinQ could be considered a low-intensity complex intervention [17] from policy to practice that provides a framework to support collaborative actions and increases the quality of care by (1) building on existing resources of the education and health care sectors, at the state and district level; and (2) enhancing their capacity to respond to mental illness in students by improving and formalising the interface between the two sectors, in particular through the appointment of district Ed-LinQ coordinators. Those coordinators, located in twelve regional Hospital and Health Services (HHS) throughout Queensland, are in charge of facilitating a strategic approach for collaboration and integration between the sectors and enabling improved access to mental health consultations, assessments, information and training opportunities, with three levers:-Strategic partnerships, including the development of collaborative interdepartmental and interagency relationships, planning and governance mechanisms;-Enhanced capacities, including workforce development strategies for the mental health, primary care and education sector staff;-Clinical guidance, including the development of information on young people’s mental health for distribution to the education sector stakeholders [18].

Following the implementation of this public intervention, an assessment of its impact was commissioned by the Queensland Mental Health Commission in 2014. The recognition of the need to assess the policy and practice impacts of complex interventions has grown in recent years [19]. Policy or practice impacts have been defined as “demonstrable changes, or benefits to products, processes, policies, and or practices, that occur after a project has concluded” [19]. These changes should be shown by measurable evolutions in practice, service delivery, commercialisation and policy.

The conditions for assessing such impacts hugely depend on the intervention implemented, the target organisation, the policy environment, the social context, the time frame for impact assessment and whether the transfer from research to policy, from research to practice or from evidence-informed policy to practice is measured. If public policy interventions in the care sector can potentially provide key findings for decision making, the number of studies that have assessed the impact of such interventions and monitor care policy strategies to determine their impact on practice (policy-to-practice impact) is low [20,21]. In addition, there is a lack of agreement on the basic components that should be assessed in impact analysis, particularly in relation to the processes involved in the adoption by the end-users of any strategy or plan.

The objective of our study was to measure the impact of the Ed-LinQ initiative from its generation as a policy strategy to its implementation in practice, as perceived by the different stakeholders involved in this process, and to acquire knowledge for the evaluation of other strategies and subsequent programs. A secondary objective was to acquire summative learning on the key components to be considered in the analysis of the process of adoption of a mental health policy strategy in the early phase of its implementation into practice.

## 2. Materials and Methods

This study was conducted by an international consortium of research centres in collaboration with the Queensland Mental Health Commission. The analysis of the impact of the Ed-LinQ initiative followed an ecological approach and used mixed methods with the collection of both qualitative and quantitative data to acquire organisational learning from a systems perspective, both for a public agency (lessons learned in the early implementation of the Ed-LINQ initiative) and international consortium (lessons learned to improve the impact analytics of the policy interventions) [22]. Research methods included surveys, documentation review (workforce development reports, Ed-LinQ coordinator reports and implementation plans), focus groups, collaboration and joint meetings with stakeholders in targeted regions (Mackay, Gold Coast and Sunshine Coast) plus in-depth interviews with Ed-LinQ coordinators and the evaluation project reference group member (representatives from government, independent and catholic education sectors). Details of the entire evaluation framework has been reported elsewhere [18]. The focus of this paper is the surveys and tools used to assess impact.

The impact of the Ed-LinQ policy strategy into practice focused on the interest in, and the adoption of Ed-LinQ at the micro (school), meso (district) and macro (state) levels. The “target audience” comprised the 1727 schools of Queensland. We focused on the process of impact of the early implementation phase or “maturity” [23] and evaluated its adoption and penetration in the target audience. As indicated in the maturity analysis of digital health tools, “Adoption” is the level to which any target organisation takes the emerging knowledge as their own (adapted from Glasgow and colleagues [24,25]). “Penetration” was defined as the proportion and significance of the organisations within the target audience that adopted the initiative (in this case, schools in Queensland). We used the Adoption Impact Ladder (AIL) to evaluate adoption and penetration. This instrument provides an ordinal measure of 7 levels of adoption by a target organisation (Table 1) [25]. The AIL reflects the increasing level of adoption from ‘0′, no adoption (the target organisation has not taken Ed-LinQ program as their own), through to ‘6′, routinisation (the target organisation has incorporated the new knowledge/Ed-LinQ into its own assessment, surveillance and monitoring systems, and has developed a plan to continue it for at least three years). The AIL usability and reliability was tested in a series of prior impact analysis studies, including (a) the adoption of a new typology for case management in Australia [26]; (b) the adoption of an action plan based on disabilities in all the public agencies of Andalusia, Spain [21]; and (c) the adoption of an international classification system for mapping health services by the regional public mental health agencies in Spain [27], and the adoption of its semiautomated Decision Support tool for planning in Andalusia (Spain) [25]. Due to the characteristics of the Ed-LinQ initiative, the measurement of the allocation alone was not considered relevant in this study and therefore levels 4 (Allocation) and 5 (Provision) of the AIL were merged into one single level. Thus, Steps 4 and 5 are represented in Level 5 of the AIL in this study (Table 1).

We created two online surveys to facilitate the assessment of the impact level as the Ed-LinQ initiative targeted multiple organizations. The first survey was directed at schools (Appendix A), while the second one was directed at child and youth mental health services (CYMHS) as well as at other mental health and health service providers (Appendix A). Following ethical approval from the Queensland Department of Education, Training and Employment, the school surveys were distributed to school principals via their respective regulatory bodies: the government schools through the Department of Education, Training and Employment, the Catholic schools through the Queensland Catholic Education Commission and independent schools through the Independent Schools Association. We targeted all government (*n* = 1272), Catholic (*n* = 269) and independent schools (*n* = 186) in the state. The e-mail invitations to participate in the school survey were sent through the state governmental agency to all school principals of Queensland public schools, Catholic schools and independent schools via the respective school’s regulatory authority. The second online survey was sent to CYMHS and other mental health and health service providers directors who forwarded them to the relevant representatives within their staff. Two follow-up reminder e-mails were sent three and seven weeks following the initial invitation via the same channels, that of their employer/executive managers. Each survey respondent provided written consent. We typified as “uninvolved” schools where a contact point opened the survey but did not reply to the invitation. It should be emphasised that the survey request was sent by the authority to whom the organisation is typically obliged to respond, in this case the school’s regulatory authority. This group was distinguished from the group of non-adopters and was not included in the descriptive analysis below.

A descriptive analysis of the surveys’ responses was used to provide a quantitative assessment of the impact of the Ed-LinQ initiative on schools and health services based on the AIL. The schools that responded positively to the survey were classified into two broad groups: (1) those where Ed-LinQ had a low adoption (corresponding to Levels 1, 2 and 3 of the AIL); and (2) those where Ed-LinQ had a moderate to high adoption (corresponding to Levels 5 and 6 of the AIL). We assessed the differences between the two broad groups of schools by studying the relationship between the adoption level as measured by the AIL and a subjective evaluation of the impact that the program had in a series of reported outcomes. A checklist for the evaluation of adoption was included in the two surveys. The questionnaire also included questions related to the perception of the initiative by the respondent, such as the overall satisfaction, the perceived improvement of the staff’s knowledge, students’ access to mental health resources, students’ attendance, and students’ performance. The schools that were rated as “no adoption” (level of impact of 0) were not included in the analysis. The responses between those with high and low adoption were then compared in order to know what factors were associated with achieving a higher impact using bivariate Pearson’s chi-squared tests (*X*^2^): statistical significance level was set at 0.05. Statistical analysis was undertaken using STATA SE 12 software. We followed the Standards for Reporting Implementation Studies (StaRI) checklist (Appendix A) to format this manuscript [28].

## 3. Results

### 3.1. Quantitative

#### 3.1.1. Impact on the School Sector (Survey 1) (Adoption and Penetration)

There were 341 government schools where a respondent opened the survey, although of these 234 (68%) did not reply, so were tagged as “uninvolved”. The response rate to the survey was 10.8% (*n* = 186). Of these, 59% (*n* = 110) were responses from government schools, 11% (*n* = 20) were Catholic schools and 30% (*n* = 56) were independent schools. The majority of them (66%) were located in the areas of Brisbane/Gold Coast/Sunshine Coast, the most urbanised areas of Queensland. No survey responses were received from the Central West, a highly remote area in Queensland (Figure 1). The majority of the contact points (persons who completed the survey) were Guidance Officers or counsellors (47%) followed by principals or heads of schools (33%) and teachers or learning support teachers (4%). Other less frequent respondents included nurses, chaplains, the dean of students, administration staff, psychologists and social workers. The schools that answered the survey tended to be large (the mean number of students attending in each school was 814); thus, from the more populated and less remote areas of Queensland (refer to Appendix A for a map of Queensland and the adoption level of Ed-LinQ across the state). There were 65% of schools that provided education for students who were five years old or less, 69% for students aged between six and nine, 94% for students aged between 10 and 14 and 77% for students over 15.

According to the AIL, 63.7% (*n* = 121) of the school representatives that answered the survey knew of the Ed-LinQ initiative. However, only 28% (*n* = 52) of the respondent schools scored over 3 in AIL (Allocation and Provision, Routinisation), which means that the school provided services, interventions and/or technologies directly related to the objectives of the Ed-LinQ initiative (Figure 2).

Out of the 68 schools that were rated “0”—no adoption of the Ed-LinQ initiative—66.7% (*n* = 45) were already participating in other mental health initiatives. Among them, 62% (*n* = 42) indicated that they would be keen to participate in more mental health-related programs. Among the schools’ respondents that were at least aware of the existence of the Ed-LinQ initiative (*n* = 118), only 33% actively disseminated information related to the initiative (Table 2).

Among the schools that were at least aware of the Ed-LinQ initiative (*n* = 118), the majority of schools (64%) were satisfied with this initiative while 25% were neutral and 11% dissatisfied.

The schools where the impact of the Ed-LinQ initiative was the highest (*n* = 52), scoring 5 or 6 on the AIL, were also the schools where more active dissemination was made by the Ed-LinQ coordinators who regularly visited the schools (*X*^2^ = 8.456, *p* = 0.004), in comparison with schools where the impact was the lowest (*n* = 67). Regarding the Ed-LinQ objectives, the schools with the highest impact perceived more improvements in coordination (*X*^2^ = 12.473, *p* = 0.002); interagency communication (*X*^2^ = 11.511, *p* = 0.003); staff’s knowledge on mental health (*X*^2^ = 17.101, *p* = 0.001); staff’s capacity to support students with mental health issues (*X*^2^ = 14.509, *p* = 0.001); and staff’s access to mental health resources (*X*^2^ = 23.625, *p* = 0.001).

However, when school staff evaluated student-related outcomes, we only found a perceived improvement in students’ access to mental health resources (*X*^2^ = 23.502, *p* = 0.001) in the schools with the highest impact. There were no significant differences between schools where the Ed-LinQ initiative had a low or high impact on the perception that the program improved students’ attendance or performance (Figure 3).

The overall penetration of the Ed-LinQ initiative in the school sector was low. Out of 1727 schools registered in Queensland’s directories at the time of the survey (target audience), only 10.8% completed it. Out of this respondent group, only 52 schools reported an AIL level of allocation/provision, and just one of these schools fully adopted the initiative as part of its routine programs. The penetration reached 3% of the target audience six years after the launch of the initiative.

#### 3.1.2. Impact on the Health Sector (Survey 2) (Adoption)

The adoption of the Ed-LinQ initiative in the health sector was assessed in a survey addressed to CYMHS and other mental health and related services. A total of 78 professionals from health services across Queensland answered the survey. Seventy (90%) provided valid and complete data that were included in the analysis, reflecting a high level of interest in the Ed-LinQ program. Among the staff who answered the survey, the majority were clinicians (39%), team leaders or directors (26%) and case managers (18%). Most of them had extensive experience as nearly half (48%) had been working in health services for more than ten years and only 8% for less than two years.

Among the professionals who completed Survey 2 (*n* = 70), 87% were at least aware of the Ed-LinQ initiative and 79% provided services, interventions and/or technologies directly related to the objectives of the Ed-LinQ initiative (Figure 4).

Ed-LinQ was viewed by 82% of the CYMHS and other mental health and related services staff as having facilitated better interactions with schools. With regard to the objectives of the program, most professionals (75%) agreed that Ed-LinQ helped build a more collaborative approach in dealing with youth mental health. The majority (68%) also agreed that it had increased the capacity of school staff to identify students in need. Finally, overall, the majority of respondents (74%) were satisfied with the Ed-LinQ initiative while 9% were neutral and 17% were dissatisfied.

### 3.2. Qualitative

In-depth interviews (each 70–120 min) were conducted by one project team member with the Ed-LinQ coordinators (*n* = 28) and expert working group (*n* = 3). Four focus groups were completed with school principals, guidance officers, student welfare officers and regional office staff, with representation from all school sectors (total participants *n* = 36). The results from the interviews and focus groups were complimentary and highlighted the positive and valued impact that the Ed-LinQ coordinator role had on the mental health of young people, and the interagency collaboration. Experts reported that referral processes in some areas were more efficient since Ed-LinQ was implemented, demonstrating positive results, with the CYMHS team receiving more Aboriginal and Torres Strait Islander students accessing their services. Areas that needed to be addressed were articulated. For example, an evaluation of the recommendations was used to customize the responses of the Ed-LinQ initiative to priority groups that have specific needs, in particular schools with higher numbers of Aboriginal and Torres Strait Islander students.

This qualitative study also identified the strengths and weaknesses of the resources allocated in the dissemination and implementation of the Ed-LinQ initiative itself. The strengths are related to the resources allocated at a strategic level, but there were less resources at the meso and micro level, including a need for ongoing reinforcement of the framework, the limited progression or deployment of the MOUs, loss of staff due to restructuring at the state-wide level, limited information on state-wide referral pathways and high-level guidance on key issues for schools, with no evidence of strategic mapping across each sector. All the qualitative results are reported in detail elsewhere [18].

## 4. Discussion

The need for better measures and indicators of accountability and monitoring for policy planning and quality assessment has been identified as a major gap in the Australian mental health system [29]. Incorporating mental health services with the existing school-based education system is a best practice integrated care approach. It involves re-orientation of the model of care and enhances the coordination of services, enabling access to services for the prevention, early intervention and treatment of a student’s mental health concerns [13,30,31]. An integrated school-based mental health approach has potential to reach a large portion of children and adolescent to promote mental health and well-being [30]. To our knowledge, this study is the first to provide a quantitative measure of the adoption of a policy initiative based on an integrated care strategy designed at the state level (micro to macro level) in Australia.

This impact assessment study quantifies the level of satisfaction, adoption and penetration within the school and the health sectors. The high impact was confined to too few schools, where 37% of the schools that answered the survey did not adopt the Ed-LinQ initiative at any level as their own, and only one fully implemented Ed-LinQ. The adoption level of the Ed-LinQ initiative was lower for schools than for CYMHS.

Those schools that responded to the survey showed a level of awareness of the Ed-LinQ initiative and perceived an improvement in student and staff access to mental health resources, better management of students with mental health issues and increased capacity of staff to support them and improved overall staff knowledge regarding mental health. They also demonstrated that staff from schools where the Ed-LinQ initiative had a higher level of adoption, felt that the objectives of the initiative in terms of service collaboration, workforce capacity building and improved services had been met. It would be helpful to identify if those schools, who were aware of Ed-LinQ, or those schools where there was higher adoption are already aware of the existing mental health standards, since awareness is recognised as a necessary predecessor to effective adoption of a new program or intervention [31].

Our findings on the level of adoption are not surprising given the low level of resources allocated to the dissemination and local implementation of the initiative identified in the qualitative analysis and the high level of need in schools. The time period between Ed-LinQ’s inception and its impact assessment should also be taken into consideration. However, the majority of schools that were aware of the Ed-LinQ initiative were responsive to knowing more about mental health programs, which highlights a significant opportunity to expand Ed-LinQ and other mental health initiatives in schools and a growing awareness of the need to tackle mental issues in children and young people.

The quantitative findings presented here mirror the perceptions of the different stakeholders of the Ed-LinQ initiative underscored in the qualitative analysis from the interviews and focus groups. It revealed that key stakeholders perceived that the Ed-LinQ initiative had had three main types of positive impacts: (1) benefits for schools, including early identification of students requiring mental health care and an increased willingness of school staff to address students’ mental health issues; (2) benefits for CYMHS, including improvements in the referral processes and reduced waiting times and crisis interventions; and (3) benefits for health education partnership and collaboration, including increased mutual support and improved working relationships. Overall, there was strong support from key stakeholders for the continuation of the initiative [18].

Previous studies of school-based mental health promotion and prevention programs have shown that interventions had an impact only if they were integrally and faithfully implemented [32]. Although the impact of the Ed-LinQ initiative was limited to several locations, it could be increased by better resourcing and a more complete and accurate implementation of the intervention. The impact analysis findings and the percentage of schools ‘uninvolved’ indicates a low level of recognition and engagement at the school-level of the Ed-LinQ initiative. The implementation of Ed-LinQ highlights the challenges of implementing a policy initiative, in particular the timely identification of issues and responses to mitigate these. Although schools can become an established platform to provide mental health services for young people, there are several challenges in successfully implementing and maintaining the transfer of evidence-based practices in school settings [33]. From attention to quality assurance to effective translation and monitoring are required for better recognition and adoption of the Ed-LinQ initiative [34,35]. A comprehensive implementation strategy is also required to engage all key stakeholders at the school level.

In Australia, the development of school-based programs for students with a lived experience of mental illness has been limited despite the recognition of its key importance [14], albeit several initiatives have been implemented in New South Wales and Victoria [36,37,38,39,40]. In particular, the School-Link Initiative in New South Wales aims to systematically formalize partnerships between schools, technical and further education colleges and mental health services, to improve the mental health outcomes for children and adolescents. The evaluation of these initiatives found similar results as in our study. They showed the crucial role of the School-Link coordinators, an improvement in the perceived adequate access to mental health services and the partnership between those services and schools. They also underscored the need for structural and curriculum changes, as well as re-orientation towards integrated care within the schools [39,41].

In our study, there was no significant association between the level of impact of the Ed-LinQ initiative as measured by the AIL and perceived improvements in the students’ performance. However, other international studies have shown that school-based interventions focusing on health and mental health promotion had clear benefits on students’ academic success [12,42].

A number of critical short- to long-term recommendations can therefore be issued to strengthen and sustain the Ed-LinQ initiative based on its initial impact assessment. Our recommendations to the Queensland Mental Health Commission and Queensland Health included the following: the development of formalised governance arrangements and guidance at the state, region and local levels, so that the key stakeholders’ roles and responsibilities are aligned and complementary; the improvement of data collection, analysis and reporting, in particular in terms of consistency, to ensure that services target the schools that are the most in need; that the best practice is identified and that the impact of the initiative is understood and quantified to inform future actions; and the clarification of the role of the Ed-LinQ coordinator and the establishment of a state-wide Ed-LinQ coordinator network, to provide coordinators with the opportunity to share resources, identify best practices and undertake joint strategic research or pilots.

This study presents useful insights into the perceived impact and level of adoption of the Ed-LinQ initiative and informs future studies. Most of the research focusing on impact analysis is still limited to the impact of research studies on future research (e.g., publication citations). Impact of policy into practice lacks a clear identification of the target organisations and target audience in a defined ecosystem (local, regional, national or international). In the future, monitoring of any policy strategy should be more systematic [43,44], and should consider a healthcare ecosystem approach [45] to enable identification of areas for improvement and a better communication of the policy’s impact to relevant stakeholders [46]. This can be particularly useful in attracting and retaining funding and localised support for the continuation of the policy initiative.

### Limitations and Strengths of the Study

There were several methodological limitations in the study. The survey’s questionnaire was completed by a single contact person representing the whole school or child mental health agency and may not reflect the perspectives of other staff who are also engaged in the implementation of Ed-LinQ or other mental health initiatives. Our analysis combined all three types of schools, and a separate analysis will be needed to adequately compare impact across school types. The government, Catholic and independents schools could differ; for example, they could be resourced differently, have different students, student bodies and faculty, different support services, and varying levels of administrative structure and support.

Unfortunately, we were not able to obtain Mail Chimp information from the Catholic and independent schools’ regulatory authorities to identify those schools who opened but did not complete the survey in these sectors. Limited contextual information available on the dissemination and implementation of the initiative also limited the interpretation of the reasons for differences in the level of adoption across schools and districts. It is important to note that some schools were already engaged in other mental health programs, which may have contributed to the lack of adoption. Likewise, we were not able to monitor the impact of external events on the evaluation, such as the effect of changes in state and national government or education policy on the deployment and roll-out of the initiative.

Our findings indicate a low response rate compared with the number of schools approached. From a traditional epidemiological perspective, the low response rate (10.8%) limits the representativity and generalisability of the findings, even though response rates for this type of survey is usually below 20% [47,48]. However, in the context of impact analysis, the response rate is key information of the appraisal of the level of awareness, particularly in an initiative promoted by the state. This information is also necessary to estimate the penetration of a policy in its target audience (in our case the school sector of Queensland). In an impact analysis this is not a limitation but a relevant finding. Similarly, the 18.2% of “uninvolved” representatives of organisations (i.e., those opening the survey questionnaire, but not responding) is also a significant finding. From an organisational learning perspective, the response rate is a relevant indicator of the adoption of the strategy and the process impact of its implementation [22]. The opening but not responding (uninvolved) thereby adds key information and knowledge to the assessment of the Ed-LinQ initiative. The group of schools who were ‘uninvolved’, demonstrated by the lack of reply to the survey and despite the direct contact by their respective education regulatory bodies and the support of the Queensland Mental Health Commission, can indeed be considered as a proxy for the dissemination and implementation challenges; a lack of recognition of the Ed-LinQ initiative; the interest of schools; and lessons learnt from the process.

There is currently a lack of a consensual methodology to assess research impact, potentially as a result of its complex nature [19,49]. However, the development of the AIL instrument to provide a quantitative ordinal rating allows for a common standard of measurement while taking into account the specificities of the project. Several factors relating to the subjective evaluation of the impact of the Ed-LinQ initiative by school staff were significantly associated with the adoption impact level measured by the AIL, which indicates external validity.

So far, the quantitative studies focusing on the policy-to-practice impact of health interventions and strategies have been mainly limited to the impact on outcomes for the individual and population [20,50,51,52,53]. A complementary measurement of the process of impact, including adoption by the target organisations and penetration in the target audience, is needed [25]; it is a necessary preliminary step in the evaluation of quality. Future studies on the impact of health policy should therefore include both the impact of the process of knowledge transfer to target organisations as well as the impact on final outcomes. These categories or domains should contribute to the development of a full taxonomy of the process of impact (input, throughput and output).

## 5. Conclusions

This assessment of the policy-to-practice impact of the Ed-LinQ initiative shows promising results, with a number of its objectives being met. While most of the mental health programs developed in schools are transient, the Ed-LinQ initiative has developed and provided support for existing partnerships through a framework that can be maintained on a long-term basis. The profiles of the low and high adopters underscored in our study may help to refine and improve policy-to-practice strategies in the future. A particular focus should be given to the dissemination, incentives to implement the program in target organisations, the mapping of potentially overlapping initiatives and benchmarking to increase the penetration of the initiative across the state.

## Figures and Tables

**Figure 1 ijerph-18-07924-f001:**
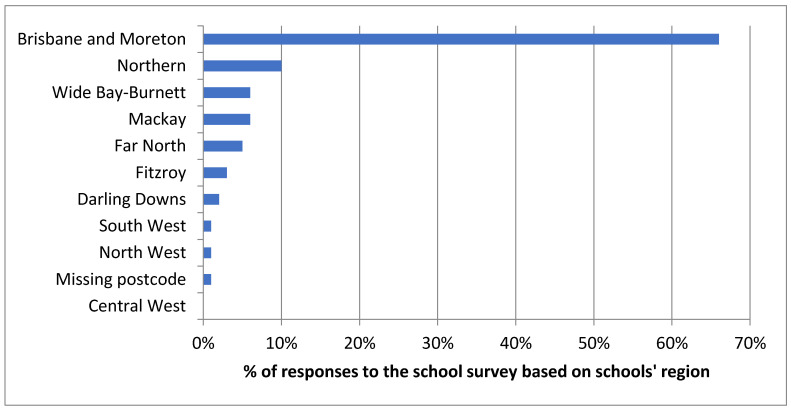
Distribution of the responses to the Ed-LinQ school survey by schools’ region in Queensland (*n* = 186).

**Figure 2 ijerph-18-07924-f002:**
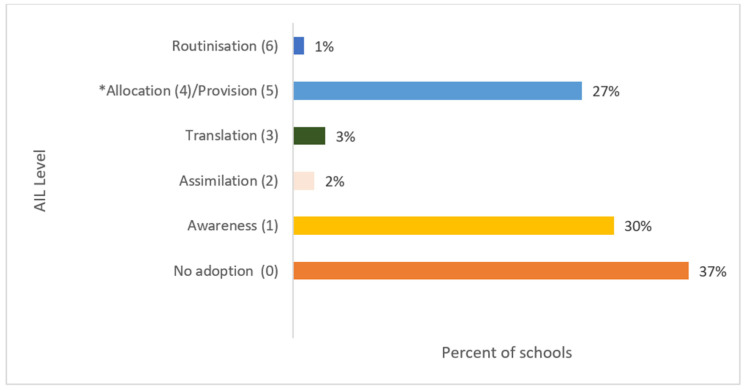
Overall adoption of the Ed-LinQ initiative in Queensland schools (*n* = 186) according to the Adoption Impact Ladder (AIL). The Allocation and Provision levels of the AIL were merged for the analysis of this initiative into a single level (Level 5). * Level 4 and 5 merged for this study.

**Figure 3 ijerph-18-07924-f003:**
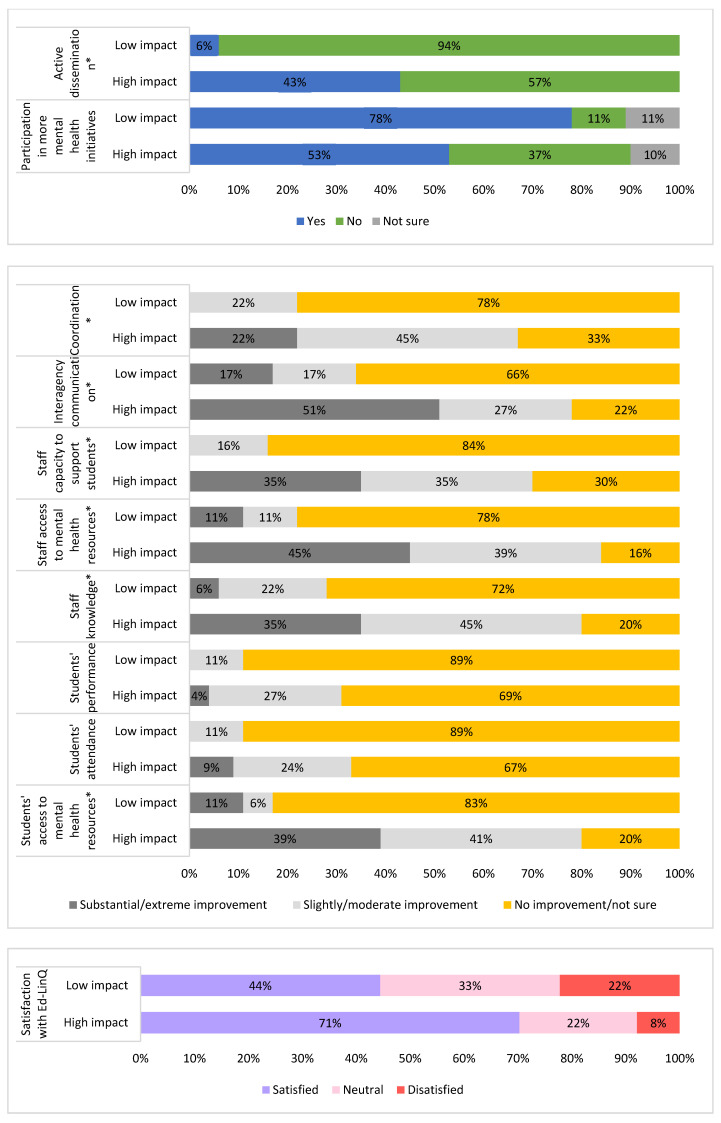
Perceptions of respondents to Survey 1 associated with the impact level of the Ed-LinQ initiative in schools (% of schools, *n* = 118). * statistically significant.

**Figure 4 ijerph-18-07924-f004:**
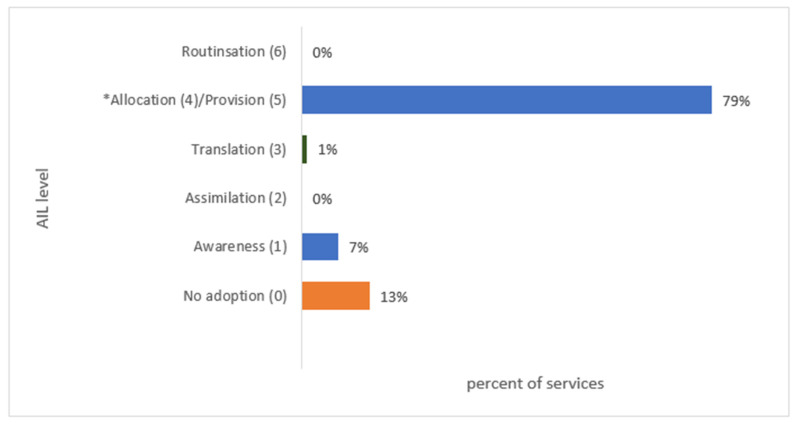
Overall adoption of the Ed-LinQ initiative in the CYMHS and other mental health and related services survey (*n* = 70), according to the Adoption Impact Ladder (AIL). The Allocation and Provision levels of the AIL were merged for the analysis of this initiative into a single level (Level 5). * Level 4 and 5 merged for this study.

**Table 1 ijerph-18-07924-t001:** Adoption Impact Ladder (AIL) adapted for the assessment of the Ed-LinQ initiative. The Allocation and Provision levels of the AIL were merged for the analysis of this initiative into a single level (Level 5).

Level of Adoption	Definition	Questions to Respondents
0. No Adoption (no impact)	The Ed-LinQ initiative was not adopted (had no impact) in the target school.	▪ Do you know the Queensland Ed-Lin Q initiative?
1. Awareness	The target schools and specific decision makers within the school/health service providers are cognizant of the Ed-LinQ initiative, have taken action to improve its knowledge on the topic or have received or provided feedback on the information delivered.	▪ If yes above, has the Queensland Ed-LinQ initiative presented at your school?
2. Assimilation	There is evidence that the target school and specific decision makers within the school have incorporated the Ed-LinQ initiative into their own existing knowledge-base and organisational strategy.	▪ Has you school discussed the possibility of engaging in the Queensland Ed-LinQ initiative? ▪ Is the Queensland Ed-LinQ initiative part of your school’s official documents, procedures, plans and reports? ▪ Has the school implemented any aspect of the Queensland Ed-LinQ initiative?
3. Conversion	The target schools have transferred the Ed-LinQ initiative into policy action in legislation, plans, policy programs, regulatory norms, and/or official indicators.	▪ Has the Queensland Ed-LinQ related initiative been incorporated in school policy and protocols? ▪ Has the school distributed information on Queensland Ed-LinQ related services for students?
(4. Allocation and) 5. Provision	The translation of the new knowledge has had an impact on financing, budgeting, funding, and/or resource allocation in the target audience.	▪ Has the school organized training for student welfare staff (e.g., guidance, counselling, nursing)? ▪ Has the school organized training of general staff on Ed-Lin Q and the policies and protocols? ▪ Provision of assessment and treatment at the school for students with mental health problems?
	Care delivery, including services, interventions and/or technologies directly related to the Ed-LinQ initiative has been made available and it is used by the target population in the school environment.	
6. Routinisation (monitoring)	The target school has incorporated the Ed-LinQ initiative into its own assessment, surveillance and monitoring systems.	▪ Has your school incorporated indicators or measures related to the Queensland Ed-LinQ initiative 9, e.g., number of referrals, number of students diagnosed with mental illness?

**Table 2 ijerph-18-07924-t002:** Perceived outcomes of the Ed-LinQ initiative on students and staff outcomes among schools that were at least aware of the initiative (*n* = 118).

Perceived Outcome	Percentage of Schools’ Respondents Reporting This Perceived Outcome
Improvement in students’ access to mental health resources	64%
Improvement in students’ attendance	27%
Improvement in students’ performance	26%
Better management of students with mental health issues	56%
Increased capacity of staff to support students	73%
Improved staff knowledge	67%
Enhancement of staff access to mental health resources	68%

## Data Availability

The data that support the findings of this study are available from the corresponding author, upon reasonable request.

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
