# Peer review of "Impact of Ed-LinQ: A Public Policy Strategy to Facilitate Engagement between Schools and the Mental Health Care System in Queensland, Australia"

_ijerph, 2021, doi:10.3390/ijerph18157924_

Round 1

Reviewer 1 Report

I have made annotated comments which are generally related to minor issues. This is a worthwhile study with high merit. Limitations beyond the control of researchers have been acknowledged.

My main concern is that I do not see any description of qualitative research methods or results. Interviews and focus groups methodology and results should be elaborated, or if that is elaborated elsewhere (reference 17), at least, they should be presented briefly in this paper too. This could be in-text or as Tables.

Author Response

We have made corrections in line with Reviewer 1 annotated comments. These are in track changes in the revised document.

Regarding the comments recommending additional information on the description of qualitative research methods and results, the following has been added

  • An overview of the implementation and dissemination is included in lines 118-129 and then reference to this has been made in the results regarding to the resources allocated (the weaknesses)
  • Description of the qualitative methods was included in section 2 materials and methods – lines 182-188
  • A brief overview of the qualitative research method results has been included in section 3 lines 402-425

Reviewer 2 Report

Thank you for the opportunity to review this interesting and thoughtful paper. You provide important insights into the complex process of translating strategy/policy into real world outcomes. Overall I think it is a very good paper. It would be helpful to provide further geographic and demographic context - the regions of Queensland are not broadly known. Perhaps the table at line 193 could be represented as a map? It would also be helpful to get a sense of Aboriginal and Torres Strait Islander population and how that might might impact on take up (eg if the program hasn't been culturally adapted it may not be relevant to school with a high Aboriginal and Torres Strait Islander population. Applying the ARIA and ICSEA or similar indices to get a sense of remoteness and disadvantage would also be helpful.

Author Response

Thankyou for your time and comments. 

We agree that additional information on the context would be helpful to the reader.

We have inserted demographic information including the percentage across the state living in regional or remote areas. (line 107-110). A map of Queensland has been inserted as an Appendix 4 and reference to this on lines 289-290. The map in Appendix 4 also reflects the remoteness, limited response and lower impact in areas outside the main urban areas.

Reference to customisation/adaptation to the Aboriginal and Torres Strait Islander school population has been included (lines 409-416).